# Regulators of homologous recombination deficiency identified by machine learning using somatic multi-omics data

Renan Valieris[1],* , Lucas Rosa[2],* , Luan Martins[2], Alexandre Defelicibus[2], Dirce Maria Carraro[2],
Diana Noronha Nunes[3], Emmanuel Dias-Neto[3], Rafael Rosales[4] , Israel Tojal da Silva[1,5]

**Homologous recombination deficiency (HRD) is a critical biomarker for guiding targeted therapies, yet the full range of somatic alterations driving HRD across cancers remains incompletely characterized. Here, we present a tumor-agnostic machine learning framework that integrates somatic multi-omics data, including copy-number variations, single-nucleotide variants, DNA methylation, and gene expression from over 8,000 patients in The Cancer Genome Atlas. Using a genome-wide mutational signature–based HRD score as ground truth, our model achieved high predictive performance and leveraged SHAP-based explainability to uncover HRD regulators beyond *BRCA1/2*. Cross-tumor analysis revealed both shared and cancer type–specific molecular determinants, whereas functional enrichment highlighted key molecular and cellular processes. These findings expand the known repertoire of HRD-associated alterations, provide a resource for mechanistic investigation, and demonstrate the potential of integrative AI approaches to improve patient stratification for HR-targeted therapies across diverse malignancies.**

## Introduction

Homologous recombination (HR) is a critical DNA repair mechanism responsible for maintaining genomic stability by accurately repairing double-strand breaks (DSBs). Deficiencies in HR (HRD) compromise this repair process, leading to genomic instability, a hallmark of cancer development ([1]). HRD tumors, including those harboring *BRCA1* or *BRCA2* mutations, exhibit heightened sensitivity to DNA-damaging agents and poly(ADP-ribose) polymerase inhibitors (PARPi). This characteristic makes HRD a critical biomarker for therapeutic approaches ([2]).

In current clinical practice, HRD assessment involves analyzing DNA lesions for germline and somatic mutations, as well as epigenetic alterations in known HR genes. In addition, three genomic scar-based biomarkers have been identified using SNP array signatures: (i) large-scale state transitions (LST), which represent chromosomal breaks between regions of at least 10 Mb ([3]); (ii) telomeric allelic imbalance (TAI), indicating unequal allele distribution in telomeric regions ([4]); and (iii) loss of heterozygosity (LOH), characterized by the distribution of LOH regions throughout the tumor genomes ([5]). Although these approaches provide predictive value for PARPi and platinum-based therapy, emerging evidence suggests that yet uncharacterized mechanisms may also play a crucial role in the context of HRD in sporadic cancers ([1], [6], [7], [8], [9]). Together, this underscores the necessity to expand our understanding of HRD-associated genes and lesions beyond currently known markers.

To address this challenge, we developed a genome-wide HRD measure using well-established mutational signatures associated with HRD ([2]). These mutational signatures, derived from global patterns of mutations in HRD tumors, provide a composite metric that serves as a robust ground truth for HRD detection ([2]). Based on the genome-wide measure, we constructed an artificial intelligence (AI)–based model capable of integrating somatic multi-omics data, including copy-number variations (CNV), single base substitutions, methylation patterns, and gene expression profiles from more than 8,000 patients in The Cancer Genome Atlas (TCGA). Our approach further incorporates state-of-the-art explainability techniques to identify additional somatic determinants of HRD beyond the well-characterized drivers such as *BRCA1* and *BRCA2*. Leveraging an integrative and explainable artificial intelligence (XAI) approach, our study not only corroborated previous findings but also unveiled novel insights, significantly enhancing our

[1]Laboratory of Computational Biology and Bioinformatics, A.C. Camargo Cancer Center, São Paulo, Brazil  [2]Clinical and Functional Genomics Group, A.C. Camargo Cancer Center, São Paulo, Brazil  [3]Division of Cancer Biology, Department of Radiation Oncology, Rutgers New Jersey Medical School, Newark, NJ, USA  [4]Departamento de Computação e Matemática, Universidade de São Paulo, São Paulo, Brazil  [5]Fred Hutchinson Cancer Center, Vaccine and Infectious Disease Division, Seattle, WA, USA

Correspondence: itojal@accamargo.org.br
*Renan Valieris and Lucas Rosa contributed equally to this work

comprehension of the molecular mechanisms that underlie the HRD phenotype, with the possibility of contributing to personalized cancer therapeutics in the near future.

# Results

### A multi-omics model for predicting HRD

We developed an aggregated HRD score, here denominated panHRD, that combines mutational signatures associated with HRD into a single score. This score incorporates the mutational signatures SBS3 (single base substitution), ID6 (indel), and CN17 (CNV), whose association with HRD has been previously described (10, 11, 12). Additionally, we incorporated TAI, LST, and LOH, which measure genome-wide scars associated with HRD (13). The aggregated score provides a broad measure of HRD-associated mutational load, which can be objectively quantified across many tumors. Next, we developed a workflow to predict the panHRD score from omics data. This workflow integrates somatic data from CNV, gene expression, protein-altering mutations, and methylation—collected from over 8,000 patients across 33 tumor types in TCGA cohort (Fig 1A), consisting in measurements of up to 60,660 genes per data layer.

Most genes are not directly associated with HRD. To reduce the dimensionality of the dataset and identify genes potentially relevant for prediction, the Boruta feature selection algorithm (14) was applied independently to each data layer, using the panHRD score as the target variable. The Boruta algorithm iteratively evaluates the importance of each feature by comparing it with that of randomized shadow features, retaining only those that demonstrate statistically significant relevance to the outcome. This procedure reduced the input data to 2,189 features across the four input layers (Fig 1B), representing ~1% of the total input features (Fig S1A–D). The reduced dataset was then used to train an XGBoost (15) regression model, optimized using cross-validation, using the panHRD score as a ground truth (Fig 1C). Our tumor-agnostic model achieved an overall AUC of 0.93 and a $R^2$ of 0.75 on a held-out test set. Finally, the interpretability of the model was performed using SHAP values (16), which highlighted the contribution of individual features to HRD prediction and revealed novel associations across the somatic multi-omics landscape (Fig 1D).

### Comparable performance of tumor-agnostic and tumor-specific models

As the full repertoire of genes and mechanisms associated with HRD remains to be fully characterized, we reasoned that integrating genome-wide mutational footprints, encompassing different genetic determinants related to HRD or their combinations, could uncover novel genes that influence HR fidelity. Hence, using an integrative ML-based approach to multi-omics somatic data, we first built a tumor-agnostic model that integrates somatic data, including gene expression, methylation, single-nucleotide mutations, and copy-number alterations from 33 different tumor types. In addition, we developed tumor-specific models, each trained exclusively on data from a single tumor type.

To evaluate the performance of these models, we calculated the area under the ROC curve (AUC) using a subset of the original dataset held out for validation. Fig 2 shows the performance differences between the tumor-agnostic model and the individual tumor-specific models, for all 33 tumors that compose TCGA cohort. Additional performance metrics for the tumor-agnostic model, a tumor-stratified breakdown, and descriptions of each tumor abbreviation can be found in Tables S1 and S2. Using our machine learning approach on a somatic multi-omics dataset, both models can reliably recover the type of cancer known to be associated with HRD at variable prevalence across cancer types (17), including BRCA (breast invasive carcinoma), ESCA (esophageal carcinoma), OV (ovarian serous cystadenocarcinoma), LUSC (lung squamous cell carcinoma), UCS (uterine carcinosarcoma), LUAD (lung adenocarcinoma), and STAD (stomach adenocarcinoma).

In addition, our analysis revealed that the tumor-agnostic model performed similar to tumor-specific models in most tumor types, demonstrating that HRD prediction is not limited to a single cancer type. In particular, the tumor-agnostic model benefited from training in multiple cohorts, which introduced regularization, minimized overfitting to the unique characteristics of individual cohorts, and facilitated the identification of more generalized and robust predictive features. Next, an additional aim of our study was to compare our tumor-agnostic model, which considers the panHRD score as the ground truth for HRD status, with the scarHRD metric (13, 18), a measure that evaluates HRD status based on the unweighted sum of TAI (4), LST (3), and LOH (5) (Fig S2). This analysis highlights the performance differences between the agnostic-tumor model and the individual tumor-specific models using scarHRD as the ground truth, while demonstrating the superior discriminatory power of our tumor-agnostic model.

### SHAP-based interpretability uncovers the somatic landscapes of HRD across tumor types

To elucidate additional genetic changes involved in homologous recombination repair, our framework leverages SHAP (SHapley Additive exPlanations) values to interpret the trained models, providing insights into the somatic data driving the model results and identifying complex patterns that traditional methods might overlook. SHAP values measure the contribution of each variable used by the model to its prediction. At the patient level, we used waterfall plots to display explanations of selected genes for HRD status in two cancer samples. In Fig 3A, we observe the absence of CNV in most genes and low SHAP contributions in the sample with HR proficiency. In contrast, the sample with genes displaying CNV exhibits positive SHAP values, highlighting the contribution of CNV in these genes to an impaired homologous recombination (Fig 3B). Next, by comparing SHAP values with the feature values across the dataset, we can calculate their correlation to interpret how the SHAP values affect the model output. A heatmap was used to present the results, effectively uncovering relationships between genes and multi-omics data across cancer types. In this analysis, rows (tumors) and columns (genes) were clustered (Fig 3C). Horizontally, the tumors were divided into two major groups. Vertically, the genes were divided into three distinct clusters: the left group

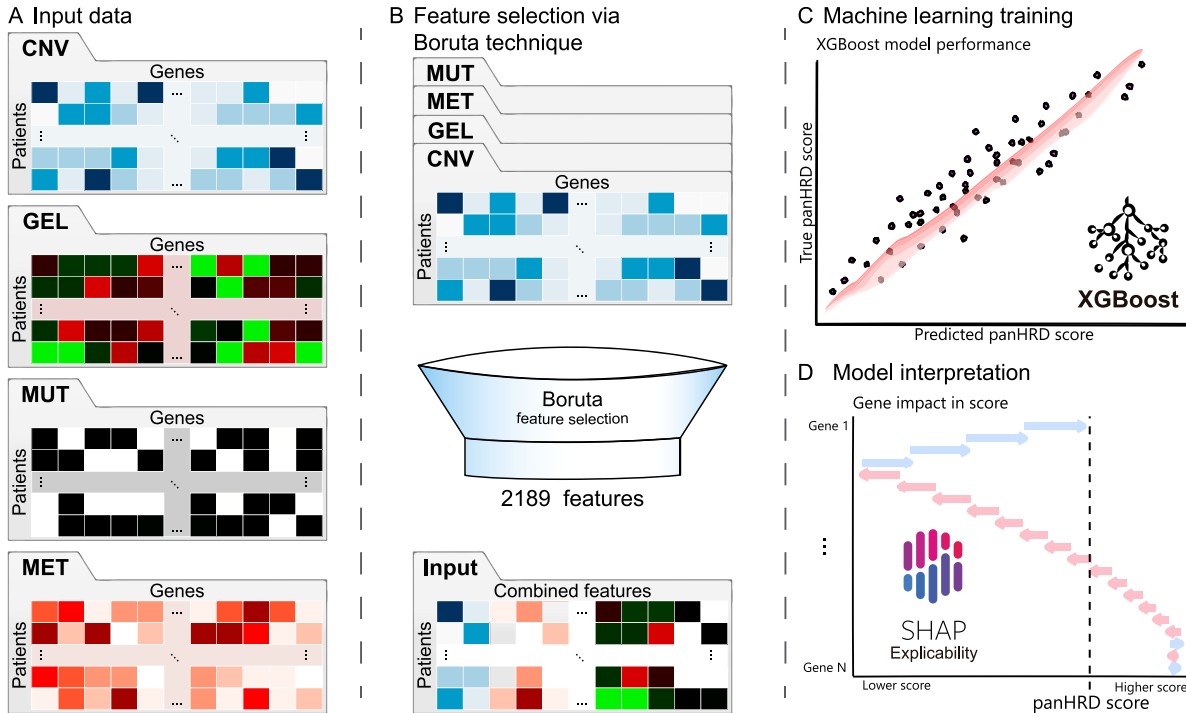

**Figure 1. Workflow of the proposed HRD prediction model.**
**(A)** Four layers of information were used as input: copy-number variation, gene expression levels, somatic mutations (MUT), and methylation (MET). **(B)** Relevant HRD features were selected using the Boruta algorithm. **(C)** Regression model was trained using the XGBoost algorithm. **(D)** Important gene features for prediction were identified using the SHAP method.

primarily represents a direct positive correlation between gene omics values (e.g., expression levels) and their corresponding SHAP values, indicating that higher feature values contribute positively to HRD prediction. The right group represents an inverse relationship, and the central group represents a tumor-specific relationship. This approach highlights the interaction between the omics data and the interpretability of the model across cancer types. The complete list of genes and their corresponding SHAP contributions for the tumor-agnostic model are provided in Tables S3 and S4.

Next, we assessed the uncertainty in hierarchical cluster analysis using a bootstrap resampling technique (19) (Fig 3D). This analysis highlights that certain tumors exhibit similar genomic profiles, suggesting underlying HRD leading mechanisms. We note UCEC, UCS, CESC, and BRCA tumors grouped in the same cluster. In our analysis, the clustering of these tumors is supported by shared molecular characteristics that go beyond anatomical and hormone-related contexts. Indeed, these tumor types exhibit HRD, often because of mutations or epigenetic silencing in key homologous recombination repair (HRR) genes (20). In addition, a comprehensive pan-cancer molecular study of gynecologic and breast cancers has identified shared genomic alterations and molecular subtypes among these tumors, indicating common pathways of tumorigenesis and DNA repair deficiencies (21). In contrast, the colorectal cancer types (referred to as COAD and READ) were found to be grouped together. Recent findings have identified putative driver mutations in genes involved in DNA damage response and repair, including those in homologous recombination pathways. Defects in these pathways can result in repair deficiencies, leading to genomic

instability because of impaired homologous recombination (22). Together, these findings provide a rationale for the observed clustering of these tumors in our analysis.

Finally, we calculated the global SHAP importance for each feature by averaging the magnitude of the SHAP values across all samples (23 Preprint). This importance score enables ranking features according to their overall contribution to model predictions. To assess how subsets of genes contribute to prediction, we leveraged XGBoost's native handling of missing values (15) and reevaluated performance on the held-out test set using only the top N features, ranked by SHAP importance, and tested increasing values for N. We then calculated Cohen's kappa (Fig S3A) and $R^2$ (Fig S3B) metrics on the test set. As a result, performance converged around the top 1–500 features, showing only modest gains up to 1,000, and changed minimally beyond 1,000 features. This pattern demonstrates that the highest-ranked SHAP features are the primary drivers of the model's predictions.

### Somatic mutation spectrum reveals unique HRD features in breast cancer

Currently, it is known that the inactivation of HRR-related genes through genetic lesions can lead to HRD (24). However, the full repertoire of somatic lesions in HR genes remains unknown. Therefore, to assess the contribution of the multi-omics interplay learned by our regression-based agnostic model (see Materials and Methods section), we used a subset of the breast cancer samples, the largest tumor cohort with annotated genomic

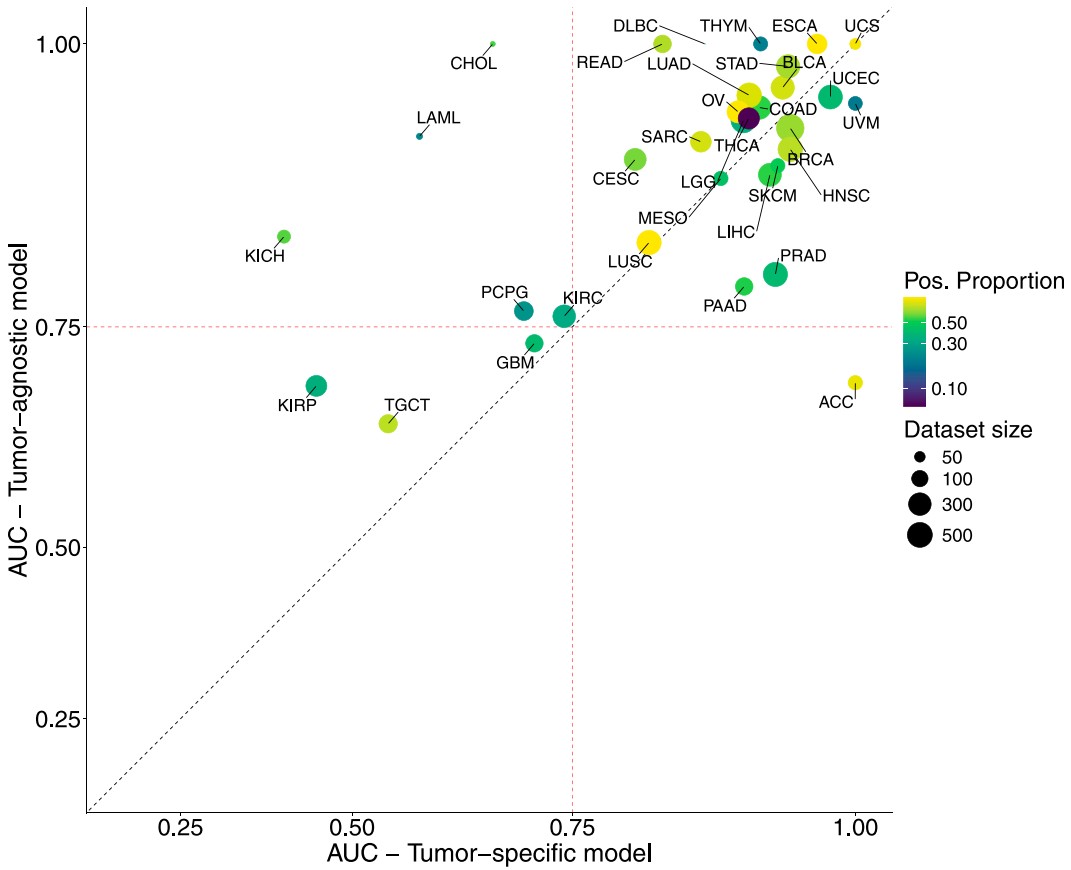

**Figure 2.   Scatter plot showing model performance on the test set.**
The AUC (area under the curve) for the model trained on all tumor types combined (y-axis, representing the AUC performance for each tumor type) versus the AUC for individual models trained on each specific tumor type (x-axis). Each point represents a different tumor type, with the size and color of the points indicating the dataset size and the proportion of positive patients (see Materials and Methods section), respectively. The red dashed lines indicate an AUC value of 0.75, whereas the black dashed line represents the line of equality (y = x).

alterations in our dataset, as a test set. The results are displayed as heatmaps (Fig 4). Samples were ordered by the panHRD score as predicted by the model (Fig 4A), displaying the top 10 genes ranked by SHAP for each omics data, including respectively CNV (Fig 4B), gene mutations (MUT, Fig 4C), gene expression levels (Fig 4D), and methylation (MET, Fig 4E). For comparison, we also included the molecular subtype information (Fig 4F) and germline status of selected HRD-related genes (Fig 4G).

Our analysis revealed two distinct patterns. (1) Positive or negative relationships: for certain genes and omics data types, we observed a clear correlation between omics values and the HRD score. For example, specific gene expression levels or mutations in genes showed trends in which increasing or decreasing values corresponded to higher HRD scores. Specifically, we found that *TP53* mutations are more frequent in samples with higher HRD score and enriched in TNBC, in line with prior reports (25). Consistent associations have been described in independent cohorts (26), and we observe a similar trend across other tumor types (Fig S4). It is important to note, however, that these results should be interpreted as an association rather than a causal relationship. Although TP53 participates in DNA damage responses and can influence homologous recombination,

*TP53* mutation alone is unlikely to cause HRD, and molecular subtype composition (Fig 4F) may contribute to the observed pattern. In contrast, lesions in *PIK3CA* were observed in HR-competent tumors and were more common in luminal A subtype tumors (27), and we observe a similar low-HRD pattern across other tumor types (Fig S5). Together, these findings recapitulate the phenotypes associated with HR status. (2) Lack of direct relationships: for other genes, we observed no consistent relationships between omics values and HRD scores. The absence of this direct correlation highlights the complexity of HRD-associated mechanisms, where contributions may be influenced by intricate, nonlinear interactions or by the broader genomic context in which these alterations occur. These findings reflect the nature of our machine learning approach, which excels in capturing both linear and more complex nonlinear relationships within high-dimensional datasets.

### Functional analysis of identified genes highlights pathway associations in HRD prediction

To gain mechanistic insights into genes selected from our agnostic model, we performed an enrichment analysis using curated gene

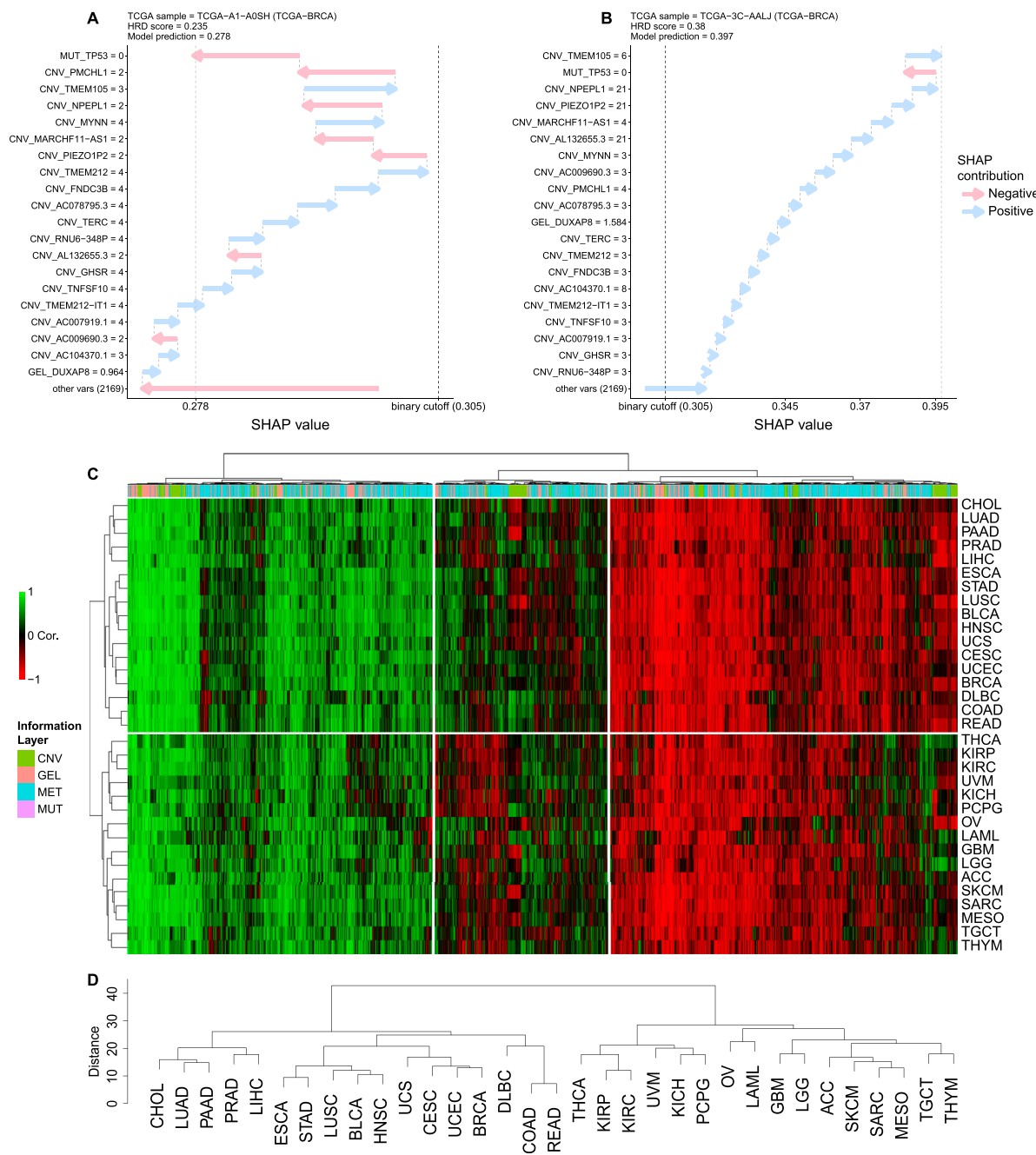

**Figure 3. Interpretability of the model features.**
**(A, B)** Waterfall plots for two breast cancer patients with negative and positive panHRD status, respectively. On the y-axis, the variables with their respective values are displayed. On the x-axis, the SHAP values corresponding to each variable are shown. **(C)** Heatmap presenting the correlation between the input feature values and the importance values assigned to these features by the SHAP technique. Genes are represented on the x-axis and tumor types on the y-axis. The color bar at the top of the graph indicates the type of information presented for each gene. Horizontally, the data are divided into three groups: the left group primarily represents a direct relationship between the gene feature and the corresponding SHAP value; the right group represents an inverse relationship; and the central group represents a tumor-specific relationship. **(C, D)** Dendrogram showing the clustering of the rows from the heatmap in panel (C), providing a more detailed view.

sets from the Human Molecular Signatures Database (MSigDB). Specifically, we used the GOBP (Gene Ontology: Biological Process), TFTGTRD (Transcription Factor Targets: Gene Transcription Regulation Databases), KEGG, and Hallmarks collections. An UpSet graph was constructed to identify common genes between gene

sets (Fig S6). Interestingly, 423 genes are not included in any collection described before, suggesting they may represent novel genes associated with homologous recombination (the complete list can be found in Table S5). In Fig 5, the enrichment analysis shows that the genetic profiles that most contribute to the

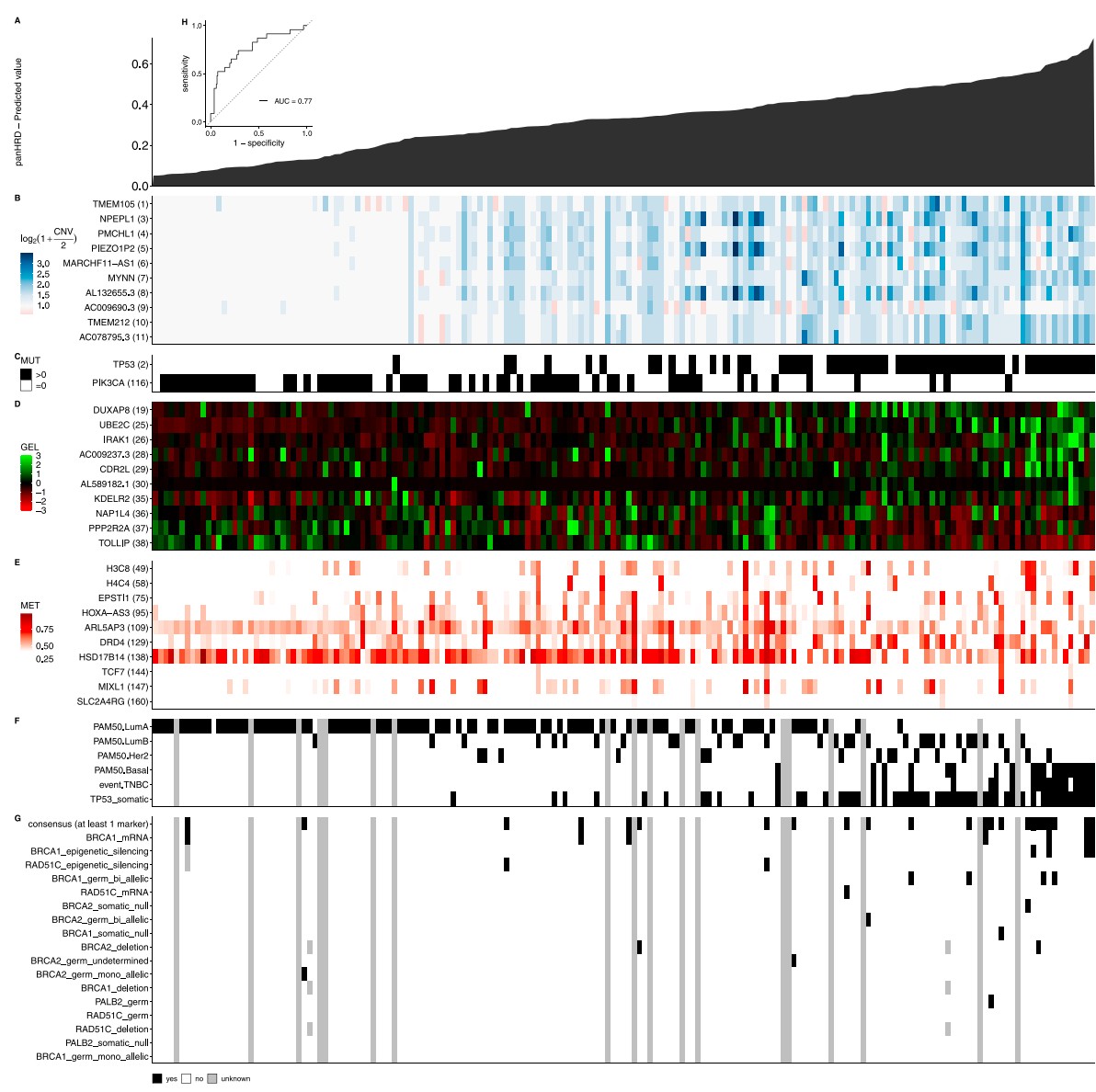

**Figure 4.  Landscape of somatic genetic lesions in predicted HR-related genes across breast samples.**

The x-axis represents the patients. In panel **(A)**, a histogram shows the levels of predicted PanHRD. Panels **(B, C, D, E)** display the top 10 genes ranked by SHAP for copy-number variation (CNV), MUT, gene expression levels, and MET, respectively. The number next to each gene name (y-axis) indicates its relative rank on the SHAP importance scale across multi-omics data. The CNV values represent deletion (0 or 1), no alteration (2), or amplification (>2), and are displayed using the transformation $\log_2 (1 + CNV/2)$. Gene expression level and MET values are presented as continuous color scaled (TPM and $\beta$ values) from red to green and from white to red, respectively. Panels **(F, G)** show overall mutation rates and clinicopathologic features, often related to breast cancer. **(G)** Panel **(H)** presents the ROC curve of panHRD as a predictor of (G) consensus.

increment of well-defined biological processes of the Hallmarks collection are gene expression (Fig 5B), methylation (Fig 5C), and copy-number alterations (Fig 5A). In addition, the analysis suggests that certain biological processes, particularly those related to DNA repair deficiencies, are similarly regulated in various tumor types. In fact, many regulatory pathways tend to exhibit coordinated alterations in response to selective pressures or specific conditions within the tumor microenvironment.

We found the enrichment of key biological processes associated with homologous recombination deficiency (HRD) or proficiency (HR-proficient), which provides insight into cellular responses to DNA damage. For example, on the interplay between DNA repair mechanisms and cell cycle regulation, HRD and elevated expression of G2/M checkpoint genes indicate a compensatory mechanism, by which cells may up-regulate G2/M checkpoint genes to delay mitotic entry, allowing additional time for DNA repair and preventing the propagation of genomic instability. This up-regulation serves as a protective response to mitigate risks associated with defective HR repair mechanisms. In contrast, HR-proficient cells possess efficient DNA repair capabilities, enabling them to promptly address DNA

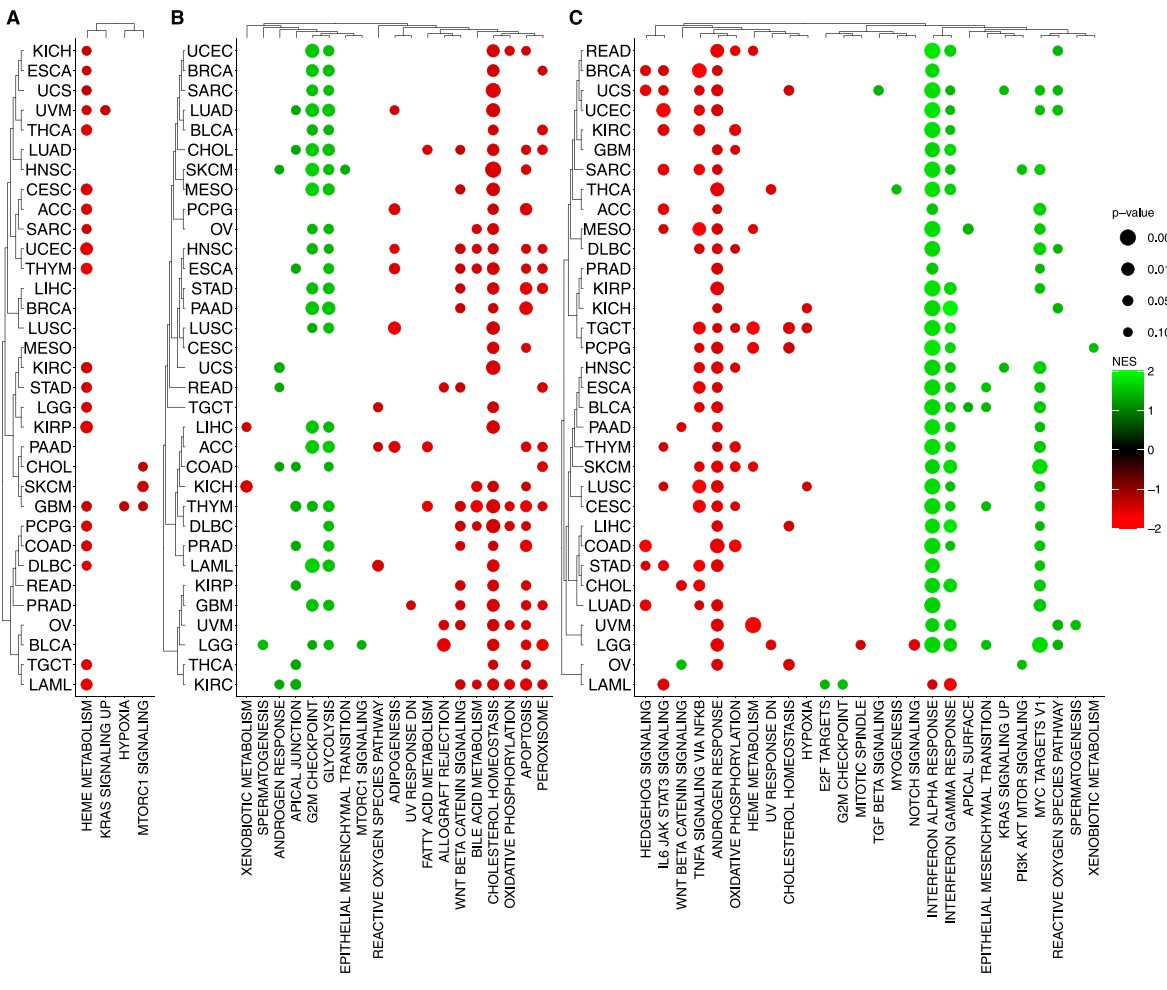

**Figure 5. Enrichment analysis of omics data.**
The bubble plot shows the enrichment of Hallmarks collection using Fast Gene Set Enrichment Analysis among the selected genes. **(A, B, C)** Each panel is associated with a different layer: (A) copy-number variation, (B) gene expression levels, and (C) Methylation. Color intensity represents the normalized enrichment score, based on SHAP importance scores and their correlation with feature values. The top pathways were selected after filtering out pathways with a *P*-value > 0.1 across all tumor types for each layer. The dot color and size indicate significance and gene enrichment magnitude, respectively.

damage with no need of prolonged cell cycle arrest (28). Therefore, these cells exhibit lower expression levels of G2/M checkpoint genes, reflecting a reduced need for stringent cell cycle control mechanisms. This efficient repair process allows for seamless progression through the cell cycle, maintaining cellular proliferation and function. Taken together, we observe that the differential expression of G2/M checkpoint genes in HRD and HR-proficient cells underscores adaptive cellular mechanisms in response to DNA repair capabilities. The enrichment analysis of the KEGG, GOBP, and TFTGTRD collections, in each of the data layers, can be accessed in Figs S7A–D, S8A–D, and S9A–D, respectively.

## Discussion

Accurate identification of HRD is crucial for better patient selection and better prognostic determination in both conventional and targeted cancer therapies (29). However, current clinical methods

for assessing HRD focus primarily on detecting germline *BRCA1/2* mutations, which are insufficient for comprehensive patient stratification. This limitation stems from the fact that HRD can arise through various mechanisms beyond *BRCA1/2* functional loss, requiring more comprehensive approaches for a better assessment (1, 2, 30). Here, we present a new AI-based model that predicts the impact of somatic other genetic alterations on HRD identification in tumor samples. We have integrated multiple data types, including gene expression, methylation, CNVs, and single-nucleotide mutations data, to evaluate the effectiveness of tumor-agnostic and tumor-specific models in predicting the HRD status. Our findings reveal that this tumor-agnostic framework achieved here is comparable in performance to other tumor-specific models, underscoring its generalizability and potential to provide actionable insights into cancers where HRD was not properly characterized.

Moreover, we employed explainability techniques, such as SHAP values, to enhance the interpretability of our findings. This

approach enabled the identification of novel HRD-associated genes and the elucidation of complex interdependencies between genomic alterations and repair deficiencies. For instance, our analysis not only corroborated the role of *BRCA1* and *BRCA2* mutations but also highlighted contributions from lesser known HRR genes to shaping the HRD phenotype. This expanded repertoire of HRD determinants aligns with recent pan-cancer analyses, which emphasize the importance of not only capturing the heterogeneity of HRD beyond BRCA-centric models (17, 24), but also retesting patients considered HR-proficient that could be indeed deficient and could benefit from treatment with platinum-based chemotherapy and PARPi therapy.

Our framework inherently integrates feature interactions and accounts for varying levels of importance across the multi-omics data, enabling the detection of subtle patterns that traditional statistical methods may overlook. The observed relationships, or lack thereof, suggest that although some genes exhibit clear predictive signals, others contribute to HRD status through more indirect or context-dependent mechanisms, potentially requiring higher order interactions for their effects to manifest. These insights reinforce the heterogeneous nature of HRD in tumor types and highlight the need for data-driven integrative approaches to fully unravel the complexities of HRD-associated pathways.

Our analysis leveraged mutational signatures to define a HRD score in terms of its mutational signature load. It is important to note that a high HRD-associated load does not necessarily imply the presence of homologous recombination deficiency. However, the two concepts are correlated, and the ability to calculate such score for a wide range of samples is what enabled the investigation of genome-wide associations that had not been previously explored.

Lastly, this study underscores the value of leveraging pan-cancer cohorts such as TCGA to validate machine learning models in oncology. The diverse and comprehensive nature of these datasets enables robust model development and evaluation across multiple cancer types, enhancing the generalizability of our findings. Future work should focus on (i) the prospective validation of this model in independent cohorts to confirm its performance across diverse patient populations and healthcare settings; (ii) its integration with emerging biomarkers, to exploit synergies between our model and novel biomarkers such as liquid biopsy-derived signatures, to further refine HRD detection; and (iii) longitudinal studies to investigate the ability of this model to track HRD status over time, providing insights into tumor evolution and adaptive treatment resistance mechanisms. Collectively, these efforts could pave the way for more precise, inclusive, and dynamic approaches to HRD detection. By improving the characterization of HRD across diverse cancer types, this framework may help generate hypotheses for treatment stratification; however, any clinical benefit remains to be established through outcome-based validation in external cohorts and prospective studies.

# Materials and Methods

## TCGA data collection

We obtained multi-omics data from TCGA cohort of 8,027 patients across 33 tumor types, consisting of gene expression data for 60,660 transcripts, methylation data for 10,986 genes, CNV for 60,623 transcripts, and somatic mutation data for 19,686 genes. The gene expression data consisted of TPM expression and were transformed with a $\log(1 + x)$ function before model fitting. The CNV data consisted of gene-level copy-number scores corresponding to the weighted median of the segments comprising the respective gene, with a score of 2 indicating no change for a diploid genome. The scores were transformed with a $\log2(1 + x/2)$ function before model fitting. The methylation data consisted of gene-level aggregated beta values from either HM27 or HM450 assays, according to the data available for each tumor type. The somatic mutation data consisted of gene-level mutation counts of mutations called by the MC3 project (31), and only mutations classified by VEP (32) as having a moderate or high impact were considered, which consists of protein-altering mutations, such as missense and nonsense mutations. Data derived from metastasis or normal samples were excluded from the dataset.

## Mutational signature estimation

Four distinct sets of mutational signatures were analyzed. Single base substitutions (SBS) were estimated from the COSMIC SBS V2 set, whereas small insertions and deletions (ID) were obtained using the COSMIC ID V3.4 set. Among these signature sets, our focus is on those linked to homologous recombination deficiency (HRD), namely, SBS3, ID6, CN17, and the scarHRD metrics (defined as the unweighted sum of LOH, LST, and TAI). The SBS3 signature represents a base substitution pattern associated with HRD (10), characterized by a flat, featureless profile. The ID6 signature is related to DNA damage repair via homologous recombination (11). CN17 is defined by segments of loss of heterozygosity (LOH) with total copy numbers (TCNs) between two and four, and heterozygous segments with TCNs between three and eight, each spanning 1–40 Mb in size (12). Finally, scarHRD metrics quantify the extent of LOH, large-scale state transitions (LSTs), and telomeric allelic imbalance (TAI), which are genomic alterations indicative of structural changes and cumulative DNA damage over time (13).

The mutational data used for both SBS and ID signature estimation were retrieved from the MC3 dataset (31), and signature activities were inferred using sigProfiler (33). CNV signatures were obtained from reference 12, and scarHRD scores were taken from reference 34 *Preprint*.

## Aggregated score for HRD

Let $\mathcal{E} = \{\mathcal{E}_k\}$, $1 \le k \le K$, be the exposures to a set of $k$ different subsets of biomarkers involved in HRD. The $k$th subset, $\mathcal{E}_k$, contains the exposures of each of i genome samples to $m_k$ possible biomarkers of type $k$. More precisely, for each $k$, let

$$\mathcal{E}_k = \{E_{k,b,i}\}, \quad 1 \le b \le m_k, \quad 1 \le i \le G, \tag{1}$$

where $E_{k,b,i}$ denotes the exposure of the genome i to the biomarker $b$ of type $k$. In this study, we consider four sets of different bio-markers for the definition of an HRD score: single base substitution mutational signatures, indel signatures, copy-number signatures, and the scarHRD (index), more precisely,

$$\begin{aligned}
\mathcal{E}_1 &= \{E_{1,b,i}\}, & b &\in \{SBS1, \dots, SBS30\}, \\
\mathcal{E}_2 &= \{E_{2,b,i}\}, & b &\in \{ID1, \dots, ID23\}, \\
\mathcal{E}_3 &= \{E_{3,b,i}\}, & b &\in \{CN1, \dots, CN21\}, \\
\mathcal{E}_4 &= \{E_{4,b,i}\}, & b &\in \{scarHRD\}.
\end{aligned} \tag{2}$$

The exposure to each biomarker in the sets $\mathcal{E}_k$, $k$ = 1,2,3, was estimated via nonnegative matrix factorization methods. The estimated exposures are, however, not necessarily comparable, especially across the different $k$ types. We consider therefore the following normalization:

$$\widehat{E}_{k,b,i} = \frac{E_{k,b,i}}{\sum_{b=1}^{m_k} E_{k,b,i}}, \quad \text{for } k = 1, 2, 3, \tag{3}$$

that is, we normalize the signature exposition in each sample, by calculating the proportion of the signature of interest by the total exposition of that sample. As such, $\widehat{E}_{k,b,i} \in [0,1]$ for all $k$, $b$, and $i$. For $k$ = 4, that is for the set constituted by the single biomarker scarHRD, we normalize with respect to the maximum observed value for all samples across the dataset:

$$\widehat{E}_{4,scarHRD,i} = \frac{E_{4,scarHRD,i}}{\max_{1 \le i \le G} E_{4,scarHRD,i}}. \tag{4}$$

We now focus on a small subset of the biomarkers in Equation (2), known to be well correlated to HRD. These are defined as follows:

$$\mathcal{E}_1 = \{E_{1,SBS3,i}\}, \quad \mathcal{E}_2 = \{E_{2,ID6,i}\}, \quad \mathcal{E}_3 = \{E_{3,CN1,i}, E_{3,CN17,i}\}, \\ \mathcal{E} = \{E_{4,scarHRD,i}\}. \tag{5}$$

Now, for the subset $k$ = 3 we combine the normalized exposures for the biomarker CN1 and CN17 as follows:

$$\widehat{E}_{3,CN1CN17,i} = \frac{\left(1 - \widehat{E}_{3,CN1,i}\right) + \widehat{E}_{3,CN17,i}}{2}, \tag{6}$$

because CN17 and CN1 are both associated with HRD, but CN1 exhibits a strong negative correlation to CN17. The set $\mathcal{E}_3$ is then redefined as $\mathcal{E}_3 = \{E_{3,CN1CN17,i}\}$. Next, we consider a regularization that takes into account the different number of genome samples of each type of tumor. Suppose there are $T$ types of tumors, and that in all $1 \le i \le G$ samples, $G_t$ are from the type t, $1 \le t \le T$. For $\varepsilon \in (0,1)$, let

$$w_{k,b,t}(\varepsilon) = \frac{1}{G_t} \sum_{i=1}^{G_t} 1\{\widehat{E}_{k,b,i} \ge \varepsilon\} \tag{7}$$

where $1\{\widehat{E}_{k,b,i} \ge \varepsilon\}$ is 1 if $\widehat{E}_{k,b,i} \ge \varepsilon$ and 0 otherwise. The quantities $w_{k,b,t}(\varepsilon)$ denote the relative fraction of samples in tumor of type $t$ with exposures $\widehat{E}_{k,b,i}$ above $\varepsilon$. The addition of different weights for each tumor type can bias the scores and make them incomparable across different tumor types. To make sure all tumors remain on the same scale, we rescale the weights such that their sum for each tumor type is the same.

$$\widehat{W}_{k,b,t}(\varepsilon) = \frac{w_{k,b,t}(\varepsilon)}{\sum_{k=1}^{4} w_{k,b,t}(\varepsilon)} \tag{8}$$

The HRD score for the ith sample of tumor of type $t$ is now computed as

$$s_{t,i}(\varepsilon) = \sum_{k=1}^{4} \sum_{b=1}^{m_k} \widehat{W}_{k,b,t}(\varepsilon) \widehat{E}_{k,b,i}. \tag{9}$$

The analyses reported here were obtained with $\varepsilon$ = 0.1. This choice gave the best results in terms of the MSE (mean squared errors) and correlation for the values predicted by the regression model.

## Feature selection algorithm

Before fitting the regression model, we aimed to reduce the number of potential predictive variables in the model. To achieve this, the Boruta (14) algorithm was employed, which is an all-relevant feature selection method. The algorithm iteratively removes features that are less relevant than random probes. It begins by creating shadow features (duplicates of all features) and shuffling their values to remove any association with the target variable. A random forest classifier is then trained on the extended dataset containing both original and shadow features. In each iteration, Boruta compares the importance score of each feature with the highest score among the shadow features. A statistical test is then performed to determine whether the importance scores of the features differ significantly from shadow features. Features whose importance scores are not significantly higher are deemed irrelevant and removed. This process is repeated multiple times, with the importance scores reevaluated in each iteration, until only relevant features remain. This ensures that only features with predictive power are retained, effectively reducing the dimensionality of the dataset, while also preserving essential information. The Boruta algorithm was applied to each omics dataset, using the HRD score as the objective, to select features relevant to the HRD score regression model.

## Regression-based model and implementation details

The model input consisted of four layers of information (gene expression, methylation, copy number, and mutations), including only the genes that were selected by the Boruta algorithm. The variables were z-scaled by subtracting their mean and dividing by their SD. For the training stage, we selected the XGBoost algorithm (15), an implementation of gradient-boosted decision trees designed for speed and performance. XGBoost is widely recognized for its efficiency in handling large datasets and its ability to improve predictive accuracy by iteratively refining models through the minimization of a loss function. We used the tidymodels framework (35, 36), for model training and hyperparameter tuning. The input dataset was initially split into a training (80%) and a test set (20%). The data were stratified by tumor type to ensure a similar distribution of tumors between the training and test set. The training set was further divided into fivefold for hyperparameter tuning using a cross-validation strategy that optimized the root mean square error metric. The final model was trained with the

optimal parameters and the entire training set. The model was evaluated using regression and classification metrics in the test set. In the latter case, the median panHRD value (0.305) across the entire cohort was used to define a cutoff point and discretize cases between HRD negative and positive samples.

### SHAP-based interpretability framework

To understand which genes were relevant for the HRD score, we used SHAP values (16), to assign the contribution of each feature to the final model prediction. SHAP is a method grounded in cooperative game theory that provides consistent and interpretable explanations for the output of machine learning models. It assigns an importance value, or "SHAP value," to each feature for a particular prediction, representing the contribution of that feature to the model's output. The SHAP values are calculated by considering the contribution of a feature across all possible combinations of features, ensuring a fair distribution of importance based on the Shapley value concept. This technique has become a widely used tool for model interpretability, particularly in complex models where understanding the influence of individual features is crucial (16). We used the treeshap package (37) to calculate the features×samples matrix of SHAP values for the corresponding test set, and the SHAP importance values for each feature, by averaging the absolute SHAP values across all samples.

### Enrichment analyses

We ranked the selected genes used by the model on their SHAP importance scores and correlation of SHAP values for their respective variables. The rank of a feature v is defined as follows:

$$Rank_v = corr(shap_v, value_v) * 10^{importance_v} \tag{10}$$

where $shap_v$ is a vector of SHAP values for each sample, $value_v$ is a vector of measure values for the respective omics information, and $importance_v$ is the aggregated SHAP importance for the respective feature v. Optimal selected genes were subjected to enrichment analyses using the fgsea package (38 Preprint). In this step, our aim was to test which gene sets were enriched in the following MSigDB collections: Hallmarks, GOBP, KEGG, and TFTGTRD for each omics data using the genes present in their respective gene sets, ranked by the $Rank_v$ values.

## Data Availability

All data used in this study were obtained from the pan-cancer web page of the GDC Data Portal (https://gdc.cancer.gov/) and are described in TCGA data collection section. The datasets generated and source code used in the analyses are available at https://github.com/tojallab/omics-reg, under the MIL license. Any additional information can be obtained from the corresponding author upon reasonable request.

## Supplementary Information

## Acknowledgements

This work was partially supported by the Fundação de Amparo à Pesquisa do Estado de São Paulo (FAPESP, #22/12991-0) and Conselho Nacional de Desenvolvimento Científico e Tecnológico (CNPq, #406002/2024-0).

### Author Contributions

R Valieris: data curation, software, formal analysis, methodology, and writing—review and editing.
L Rosa: software and formal analysis.
L Martins: software and visualization.
A Defelicibus: data curation and software.
DM Carraro: data curation and formal analysis.
DN Nunes: formal analysis and writing—review and editing.
E Dias-Neto: formal analysis and writing—review and editing.
R Rosales: formal analysis, methodology, and writing—original draft, review, and editing.
IT da Silva: conceptualization, supervision, funding acquisition, methodology, project administration, and writing—original draft, review, and editing.

### Conflict of Interest Statement

The authors declare that they have no conflict of interest.

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
