## [Reviewer comments · Life Science Alliance]

Regulators of homologous recombination deficiency identified by machine learning using somatic multi-omics data

Renan Valeris, Lucas Rosa, Luan Martins, Alexandre Defelicibus, Dirce Carraro, Diana Nunes, Emmanuel Dias-Neto, Rafael Rosales, and Israel da Silva

DOI: <https://doi.org/10.26508/lsa.202503531>

Corresponding author(s): Israel da Silva, AC Camargo Hospital

Review Timeline:

Submission Date:	2025-10-12
Editorial Decision:	2025-10-14
Revision Received:	2025-10-17
Editorial Decision:	2025-10-22
Revision Received:	2025-10-28
Accepted:	2025-10-31

Scientific Editor: Tim Fessenden

Transaction Report:

Please note that the manuscript was previously reviewed at another journal and the reports were taken into account in the decision-making process at *Life Science Alliance*. Since the original reviews are not subject to Life Science Alliance's transparent review process policy, the reports and author response cannot be published.

October 14, 2025

Re: Life Science Alliance manuscript #LSA-2025-03531-T

Israel Tojal da Silva
Fundacao Antonio Prudente

Dear Dr. da Silva,

Thank you for transferring your manuscript entitled "Machine learning model of somatic multi-omics data reveals HRD regulators beyond BRCA-centric models" to Life Science Alliance. In accordance with our offer conveyed in the decision letter from another journal, we invite you to submit a revised manuscript addressing the reviewer points.

Thank you for this interesting contribution to Life Science Alliance. We are looking forward to receiving your revised manuscript.

Sincerely,

B. MANUSCRIPT ORGANIZATION AND FORMATTING:

October 22, 2025

RE: Life Science Alliance Manuscript #LSA-2025-03531-TR

Dr. Israel Tojal da Silva
AC Camargo Hospital
Computational Biology and Bioinformatics
Tagua, 440
Sao Paulo, Sao Paulo 01508-010
Brazil

Dear Dr. da Silva,

Thank you for submitting your revised manuscript entitled "Machine learning model of somatic multi-omics data reveals HRD regulators beyond BRCA-centric model". We have evaluated the revised manuscript without further reviewer input. We feel this work has improved substantially according to the reviewer points stipulated in our previous decision letter.

Pending these final text changes, and the formatting requirements noted below, we would be happy to publish your paper in Life Science Alliance.

- To improve clarity and with a broad readership in mind, we suggest changing the title to: "Regulators of homologous recombination deficiency identified by machine learning using somatic multi-omics data."
- Please upload your main manuscript text as an editable .doc file.
- It is recommended to exclude figures from the manuscript text and upload them separately.
- Please upload all figure files individually, including the supplementary figure files; all figure legends should only appear in the main manuscript file.
- Please add a Summary Blurb / Alternate Abstract in our system.
- Please add the X and Bluesky handles of your host institute/organization as well as your own and/or one of the authors in our system.
- Please incorporate any points from the Conclusion section into the Discussion; we only allow a Discussion section.
- Please add Author Contributions and Acknowledgments sections to your main manuscript text.
- Please add your main, supplementary figure, and table legends to the main manuscript text after the References section.
- Tables can be included at the bottom of the main manuscript file or sent as separate files in editable .docx or .xlsx format.
- Supplementary figures should be labeled as Figure S1, Figure S2, etc. Tables also: Table S1, Table S2, etc.
- Please add callouts for Figures 4A, F, G; 5A-C; S1A-D; S4A-D; S5A-D; S6A-D; and S9A-B to your main manuscript text.
- The cancer type abbreviations are never spelled out in the text. Please include the full names of cancer types, for instance as a new column in Table 2, and point the reader to this list in the text.

LSA now encourages authors to provide a 30-60 second video where the study is briefly explained. We will use these videos on social media to promote the published paper and the presenting author (for examples, see <https://docs.google.com/document/d/1-UWCfbE4pGcDdcgzcmiuJl2XMBJnxKYeqRvLLrLS08s/edit?usp=sharing>). Corresponding or first-authors are welcome to submit the video. Please submit only one video per manuscript. The video can be emailed to contact@life-science-alliance.org

A. FINAL FILES:

B. MANUSCRIPT ORGANIZATION AND FORMATTING:

Thank you for your attention to these final processing requirements. Please revise and format the manuscript and upload materials as soon as you are able.

Sincerely,

October 31, 2025

RE: Life Science Alliance Manuscript #LSA-2025-03531-TRR

Dr. Israel Tojal da Silva
AC Camargo Hospital
Computational Biology and Bioinformatics
Tagua, 440
Sao Paulo, Sao Paulo 01508-010
Brazil

Dear Dr. da Silva,

Thank you for submitting your Research Article entitled "Regulators of homologous recombination deficiency identified by machine learning using somatic multi-omics data". We note the text added to the results section, which we hope you agree better introduces your method for a broad readership. It is a pleasure to let you know that your manuscript is now accepted for publication in Life Science Alliance. Congratulations on this interesting work.

DISTRIBUTION OF MATERIALS:

Again, congratulations on a very nice paper. I hope you found the review process to be constructive and are pleased with how the manuscript was handled editorially. We look forward to future exciting submissions from your lab.

Sincerely,
